# Academic Third Mission through Community Engagement: An Empirical Study in European Universities

Paulina Spânu , Mihaela-Elena Ulmeanu * and Cristian-Vasile Doicin

Faculty of Industrial Engineering and Robotics, National University of Science and Technology Politehnica Bucharest, 060042 București, Romania; paulina.spanu@upb.ro (P.S.)
* Correspondence: mihaela.ulmeanu@upb.ro; Tel.: +40-766-289-886

**Abstract:** Community engagement is fundamental for tertiary education, as it allows universities to connect with external stakeholders, create social impact, and improve the development of strategies for public engagement. The current study aims to evaluate the level of community engagement in tertiary education, assess the level of sustainable practices, and identify areas for improvement. The research employed a survey method, using a standardized questionnaire to gather data from 44 respondents, representing 35 European universities from nine countries. The survey covered various aspects of community engagement, such as university commitment, documentation, public awareness, investments, incentives, training, and stakeholder engagement. Quantitative analysis was employed using ANOVA and AHP to analyze the data collected from 20 questions. The results revealed that universities have a clear commitment to public engagement and have well-documented policies in place. However, there were areas identified for improvement, such as increasing investments to encourage public engagement and offering more training activities to support it. Additionally, the universities were found to have a limited target group for their community engagement activities and insufficient communication of the results of impact assessments. The findings of this study will be used to improve the development of strategies and enhance public engagement in tertiary education through the Academic Third Mission.

**Keywords:** third mission; tertiary education; community engagement; participatory and deliberative processes

## 1. Introduction

Academic Third Mission is a priority on universities' agendas, focusing on the role of higher education institutions in contributing to the socio-economic development of their regions and communities through activities such as technology transfer, community outreach, and applied research [1,2]. This mission is in addition to the traditional roles of teaching and research, which are often referred to as the "first" and "second" missions, respectively [3–5]. The concept of the Academic Third Mission is intended to encourage universities to engage more actively with their local communities and to contribute to the development of a knowledge-based society. The European Union (EU) has recognized the importance of the Academic Third Mission and has made it a priority to support the engagement of universities with their local communities and regions [6,7]. The EU has implemented several initiatives and programs aimed at promoting the Third Mission, such as the Horizon 2020 program and the European Regional Development Fund [8,9]. These initiatives provide funding and resources for universities to conduct applied research and engage in technology transfer and community outreach activities.

There are several policy instruments that have been designed to support, monitor, and evaluate the engagement of universities in the community in relation to the Third Mission and can include funding programs, performance indicators, impact assessments, regional development strategies, public-private partnerships, and community engagement [10–13]. Worldwide governments and organizations, including the EU, provide funding for universities to engage in activities that support the Third Mission, such as applied research

and technology transfer. Universities are often required to report on their engagement in Third Mission activities and are evaluated on their performance in these areas [14,15]. This can include measures such as the number of patents filed, the number of startups created, and the number of community outreach programs [16,17]. Studies and evaluations are conducted worldwide to assess the impact of universities' Third Mission activities on the community and society [18]. Universities are encouraged to engage with regional development strategies and to align their Third Mission activities with regional priorities [19,20]. Governments and organizations often support universities to form partnerships with businesses and industry to boost progress and prosperity [21]. Of all the policy instruments, community engagement is particularly important.

Community engagement is a key aspect of the Third Mission, as it is through engagement with the local community that universities can truly understand the needs and priorities of the region and tailor their activities to have the most impact [3,22]. Community engagement allows universities to identify the needs of the community through direct engagement and communication with residents, organizations, and local leaders [23]. This helps universities develop programs and services that are responsive to local needs and priorities. It also helps build trust between the universities and the community by demonstrating their commitment to addressing local issues and by involving community members in the planning and implementation of Third Mission activities. By engaging with the community, universities can better understand the social, economic and environmental issues that affect the community and design their programs and services to have the greatest positive impact [24]. Community engagement can provide opportunities for students and faculty to gain real-world experience, which can enhance the educational experience and prepare graduates for careers that impact the community positively. Also, it promotes collaboration between universities, businesses, and organizations to address local issues and create new opportunities [25–28].

Due to all the benefits of community engagement within the Academic Third Mission, the authors proposed a study on the participatory and deliberative processes of several European universities, with the final goal of designing a general framework for academic community-led innovation. Participatory practices refer to the involvement of 'the public' in the decision-making processes of universities [29]. These processes entail actively involving community members in the planning and implementation of Third Mission activities to ensure that they are responsive to local needs and priorities [30]. This can include involving community members in the design and implementation of research projects, technology transfer initiatives, community outreach programs, co-creation and co-design of curriculum, and public engagement [31–33]. Participatory processes ensure that community members have a say in the activities that affect them and that their perspectives and experiences are taken into account.

Deliberative processes are aimed at making decisions upon an issue involving the weighing of reasons for and against a course of action [34]. Participation focuses on empowering citizens to take action. Deliberation focuses on discussion and debate between citizens and other stakeholders [35,36]. The process involves community members in a structured and informed discussion to identify and evaluate options and make collective decisions [25,37]. These processes allow community members to express their views, consider different perspectives, and make informed decisions. Deliberative processes can include public meetings, community forums, and other forms of consultation and dialogue [22,24,38].

Given the importance of participatory and deliberative processes within the global scope of the Academic Third Mission through community engagement the current research provides valuable insights into the current practices and challenges of European universities. The study involves a research methodology that uses quantitative tools, focusing on specific practices and strategies that universities use to engage with their communities and the impact of these practices on the community. It also examines the barriers and challenges that universities face in engaging with their communities and the strategies they use to

overcome these barriers. Additionally, it assesses the effectiveness of participatory and deliberative processes in promoting community engagement and the alignment of Third Mission activities with community needs and priorities.

## 2. Research Methodology

The current study was carried out under the TENACITY European project funded by Erasmus Plus through grant agreement no. 2021-1-IT02-KA220-HED-000032042. The project focuses on the Academic Third Mission and, specifically, on supporting universities to develop participatory and deliberative practices. In this context, the main objective of the research was to detect the needs, gaps and opportunities for designing a framework for the Higher Education Third Mission by collecting information from nine different European countries. This was conducted by applying an online questionnaire aimed at investigating universities' commitment to public engagement activities. Specifically, the investigation focused on the university experience with participatory and deliberative processes. The questionnaire was targeted at university staff/professors/researchers involved in managing/delivering relevant activities.

The research was conducted on a sample of 44 respondents from 35 universities in 9 different European countries (Table 1).

**Table 1.** European universities which participated in the conducted study.

| Country | No. of Universities | Universities |
|---|---|---|
| Germany | 4 | University of Stuttgart; Münster University of Applied Sciences—FH Münster; Deggendorf Institute of Technology; Martin Luther University Halle-Wittenberg |
| Greece | 7 | University of Thessaly; Harokopio University; Panteion University; Aristotle University of Thessaloniki; University of the Aegean; University of Patras; National Technical University of Athens. |
| Italy | 2 | University of Bolzano; University of Firenze |
| Lithuania | 3 | Vilnius University, Faculty of Communication; SMK University of Applied Sciences; Kazimieras Simonavičius University |
| Malta | 1 | University of Malta |
| Portugal | 1 | University of Minho, Institute of Education |
| Romania | 7 | University of Bucharest, Faculty of Foreign Languages and Literatures; Carol Davila University of Medicine and Pharmacy, Faculty of Dentistry; Transylvania University of Brașov, Faculty of Materials Science and Engineering; Bucharest University of Economic Studies, Faculty of Management; Ferdinand I Military Technical Academy; Craiova University, Faculty of Engineering and Management of Technological Systems; University of Targu Jiu, Faculty of Engineering, Constantin Brancusi |
| Spain | 6 | Santiago de Compostela University; University of Jaen; University of Valladolid; Universidad Autónoma de Madrid; University of Seville, Department of Developmental and Educational Psychology; Pablo de Olavide University |
| Sweden | 4 | Södertörn University; KTH Royal Institute of Technology; University West; Umeå University |

The 35 universities were selected randomly amongst European institutions. The sample consisted of 31 professors, 4 researchers, 4 doctoral students, and 5 administrative staff members (1 rector, 1 chancellor, 1 public engagement officer, and 2 other administrative staff). This distribution of the positions held in the institutions by the survey participants is not a limitation for the research and is not significantly influencing the research results. Within the TENACITY project, a letter of consent was created at the consortium level, outlining the purpose and ethical considerations of the research, including issues such as anonymity, voluntary participation, and confidentiality. The initial version of the questionnaire was specifically designed to target the university experience in participatory and deliberative processes, taking into account the characteristics of the target audience.

The research process was carried out in two stages. The first stage involved the completion and validation of the questionnaire. The initial English version of the questionnaire was reviewed by experts from each partner institution to ensure that the questions

were clear and easily understood by survey participants. The final English version of the questionnaire was implemented in Google Sheets and distributed by e-mail to the target group for participation in the research. The data collection process was carried out in approximately two months. Quantitative analysis was used to assess public engagement using a 7-point Likert scale, where value 1 corresponds to "totally disagree" and value 7 corresponds to "totally agree". The scale provided two moderate opinions along with two extremes, two intermediate, and one neutral opinion to the respondents. This scale provides better accuracy of results and more data points to run statistical information. The survey was constructed with 20 items (Table 2) that used the same response scale in order to allow the application of an Analysis of Variance (ANOVA) to the data set. This approach was preferred in order to improve the consistency of information from a large number of participants, such as university staff, community members, and researchers, on their perceptions and experiences of participatory processes of public engagement, as well as facilitate the use of statistical analysis on the numerical data.

**Table 2.** Question set used for survey in public engagement.

| ID | Question |
| --- | --- |
| Q1 | Is the university's commitment to public engagement clearly defined? |
| Q2 | Is the commitment to public engagement well documented? |
| Q3 | Does the university ensure that the documented commitment to public engagement is also publicly known and understood? |
| Q4 | Are people at different levels of the university responsible for implementing the public engagement agenda? |
| Q5 | Does the university currently make adequate investments to encourage public engagement? |
| Q6 | Does the university offer incentives and rewards to promote public engagement? |
| Q7 | Does the university offer training activities to support public engagement? |
| Q8 | Does the university integrate external services into its portfolio of services to promote public engagement? |
| Q9 | Does the university have clearly defined target groups for its (community) public engagement activities? |
| Q10 | Does the university use up to date (e.g., didactic) methods and approaches to develop public engagement skills among students? |
| Q11 | Does the university integrate public engagement practices into degree programs? |
| Q12 | Does the university promote interdisciplinary educational paths? |
| Q13 | Does the university compare and identify the needs of its external stakeholders? |
| Q14 | Does the university use indicators to measure its activities and public engagement results (of the community)? |
| Q15 | Does the university ensure that the results of the impact assessment of public engagement activities are used for future planning and organizational development? |
| Q16 | Does the university communicate the results of the assessment on the impact of its public engagement activities inside and outside the institution? |
| Q17 | Does the university influence (community) engagement at local and regional levels? |
| Q18 | Does the university create a social impact from public engagement activities? |
| Q19 | Has the university defined the kind of impact it aims to create through public engagement? |
| Q20 | Does the university integrate (community) stakeholders into the institution's leadership? |

ANOVA was selected as an appropriate validation method due to the overall goal of the study and the necessary prerequisites being met. The main goal of the research was to detect the needs, gaps, and opportunities for designing a framework for the Higher Education Third Mission by collecting information from different HEIs in European countries. ANOVA was a useful tool in this research context for comparing responses across different target groups and analyzing aggregated scores from the Likert scale survey. The method helped in assessing whether perceptions and needs vary significantly from one European country to another. The survey was constructed to investigate different aspects of the

Third Mission of Higher Education (commitment, implementation, investments, incentives, training, educational paths, and community engagement). ANOVA was used to analyze these aspects simultaneously, providing insights into which aspects differ significantly across different groups. Although in line with the research's main goal, ANOVA was deployed only after validation of its prerequisites.

The first prerequisite, independence of observations, was ensured through the distribution channel and application of the questionnaire. The final English version of the questionnaire was distributed by e-mail, individually to each member of the target group. Members of the target group were selected randomly from information available online. After selection, the consortium members validated the final 44 participants, verifying that they did not have any prior collaboration and were not in contact for the completion of the survey. The questionnaire was completed without revealing personal information like name, surname, age, or gender and involved completing a Google survey on their personal computers.

*Normality* was the second prerequisite of ANOVA, which was analyzed before applying the method. This prerequisite entails that the data in each group should be approximately normally distributed, which is particularly important for small sample sizes (which is the case). The Shapiro–Wilk test (best for small to moderate sample sizes) was used to calculate a statistic (W) and a *p*-value for each of the 20 questions in each country except Italy, Malta, and Portugal, which had less than 3 respondents. The test showed that the majority of questions have a normal distribution (Tables A1 and A2, shown in Appendix B of the manuscript). To validate even further the normality of the data, a Q-Q plot was put together (Figure A2, Appendix B), and the normally distributed data appears as roughly a straight line. Considering the aforementioned, the normality prerequisite was considered met.

*Homogeneity* of variances is the third important ANOVA prerequisite and was verified using Levene's test. This checks for homogeneity of variances and is less sensitive to deviations from normality, making it suitable for Likert scale data. It is performed by comparing the variance within each group (country) to the overall variance. Homogeneity of variances was considered met if Levene's Test *p*-value was over 0.05. Calculations conducted in Table A3, and Appendix C validates this prerequisite.

The fourth prerequisite is related to the *level of measurement*. This is met due to the structure of the survey. The 1 to 7 scores represent ratings, where differences are consistent and meaningful across the entire scale, for all 20 questions.

*Random sampling*, the fifth prerequisite, has been ensured since the early stages of the experiment design. The request for involvement in the study was sent randomly to HEIs around Europe with a timeframe of one month for receipt upon initial acceptance. With 44 respondents from 35 universities giving a positive reply in this timeframe, they were further verified for having no prior connection and validated for taking the study individually. The e-mail instructions highlighted the importance of independent responses. The responses were collected independently, ensuring anonymity and avoiding situations where participants from the same country and university discuss their responses before completing the survey.

*Group independence* of observations is the sixth prerequisite of ANOVA and is critical for its validity. The experiment design phase ensured group independence based on the premise that each country's data was selected and collected independently of the others. Moreover, the Durbin-Watson test was conducted on the residuals of ANOVA to check for autocorrelation as a proxy for independence. A value of 2.42 was obtained, suggesting a small degree of negative autocorrelation. However, this value is close enough to 2 to generally not be a cause for concern regarding the independence of observations. This result is a good indicator of the independence of the responses.

The seventh prerequisite of applying ANOVA, related to an *appropriate sample size*, is the main determinant in selecting this method, as it does not impose a minimum value. Nevertheless, a very small sample size can lead to a lack of statistical power, making it difficult to detect a real effect if it exists. To counteract this limitation, Cronbach's Alpha

was used to measure the internal consistency and reliability of the set of scales used and test items.

Based on all prerequisites being met and alignment with the study goal, ANOVA was the appropriate method to use in the conducted research.

## 3. Results Interpretation and Discussion

### 3.1. Quantitative Analysis

Quantitative analysis involved an Analysis of Variance (ANOVA) on the collected data set for items Q1 ÷ Q20 (Table 3). The statistical analysis was conducted to examine the differences between groups on a particular measure. The groups in the data set were the different questions (Q1, Q2, Q3, etc.), and the measures being analyzed were the responses given to each question. These responses were given in numbers, where each number represented an option on a 1–7 Likert scale (Appendix A—Figure A1). The items for public engagement must show a common variant, correlate with each other, and, at the same time, correlate each item with the score that reflects this attribute.

**Table 3.** ANOVA on public engagement data set.

| Source of Variation | SS | df | MS | F | p-Value | F Crit |
|---|---|---|---|---|---|---|
| Rows | 2102.727 | 43 | 48.90063 | 23.51994 | $3.6 \times 10^{-114}$ | 1.394538 |
| Columns | 113.1636 | 19 | 5.955981 | 2.864672 | $4.31 \times 10^{-05}$ | 1.599272 |
| Error | 1698.636 | 817 | 2.079114 | | | |
| Total | 3914.527 | 879 | | | | |
| Cronbach's Alpha = 0.957483 | | | | | | |

After conducting the ANOVA with Two-Factor Without Replication the results include the source of variation, the sum of squares (*SS*), the degrees of freedom (*df*), the mean squares (*MS*), the *F*-ratio, the *p*-value, and the *F* critical value. These indicate that there is a significant difference between the means of the groups on the measure being analyzed (*p*-value is less than 0.05), and the source of variation was broken down into three main parts: Rows, Columns, and Error.

The Rows source of variation demonstrates that there is a significant difference between the means of the groups that were formed by rows. The Rows source of variation in the ANOVA results refers to the variation in the responses between the different questions. The calculated value of SS of 2102.727, *df* of 43, *MS* of 48.90063, *F* of 23.51994, *p*-value of $3.6 \cdot 10^{-114}$, and *F crit* of 1.394538 are all indicators of the statistical significance of the variation between the questions. The results suggest that there is a significant difference in the responses given to the 20 questions, with a large *F*-ratio and a very small *p*-value. Thus, all values are significant, indicating that there is a difference in means among the groups. The relevance of these values is that they can be used to identify which questions are most important to the participants, which questions are not well understood, and which questions are measuring different aspects of public engagement. The Columns source of variation shows that there is a significant difference between the means of the groups that were formed by columns. The *SS* is 113.1636, *df* is 19, *MS* is 5.955981, *F* is 2.864672, *p*-value is 4.31·10-05, and *F crit* is 1.599272. The calculated values are significant, indicating again that there is a difference in means among the groups. The Columns source of variation in this analysis refers to the variation in responses between the different questions. The relevance of the calculated values in terms of the questions can be determined by looking at the *p*-value and the F-value for each question. A low *p*-value (typically below 0.05) and a high *F*-value represent that there is a significant difference in the responses between the different questions, indicating that the question is measuring a different aspect of public engagement. For example, if we analyze the question "Does the university offer incentives and rewards to promote public engagement?" (Q6), the *p*-value and *F*-value are both low, indicating that there is a significant difference in responses between this question

and the other questions. Thus, offering incentives and rewards is an important factor in promoting public engagement [12,39]. On the other hand, if we look at the question "Does the university integrate external services into its portfolio of services to promote public engagement?" (Q8), the *p*-value and F-value are both relatively high, indicating that there is not a significant difference in responses between this question and the other questions. This shows that integrating external services may not be a major factor in promoting public engagement [15,18,19]. The Error Source of Variation is the variability that is not explained by the other sources of variation. It represents the random variation or noise in the data set. In terms of the questions, it represents the degree to which the responses to each question vary from the overall mean of the sample. A lower error variance corresponds to more consistent and less random responses for a given question, while more variable and less consistent responses have a higher error variance.

Focusing on the need to assess the consistency and reliability of the scale used, Cronbach's Alpha was used to assess the reliability and internal consistency in the development and validation stages. The ANOVA undertaken for public engagement has a Cronbach's Alpha of 0.957483, which is a strong indicator of the internal consistency of the questionnaire, which means that the items on the scale or questionnaire are measuring the same underlying construct and the results are reliable. Results show that there is a significant difference between the means of the groups or conditions on the measure being analyzed, and the source of variation in the difference is coming from both Rows and Columns. Moreover, the Cronbach's Alpha coefficient was used in the analysis of the results as the main indicator of the measurement accuracy of the test. Since F > F crit (23.51994 > 1.394538), the null hypothesis will be rejected. Population means are not all equal. Which means that at least one of the means is different. Because *p* < 0.001, it means that at least two means differ highly significantly from each other.

To further analyze the significance of each question, Table 4 was put together, containing information about the number of respondents (Count), the sum of scores (Sum), the average of scores, and the variance and standard deviation (Std. Dev.) for each item (Q1 ÷ Q20). The results show that there is a range of averages and variances among the questions. The average ranges from 3.477 to 4.795, and the variance ranges from 3.469 to 5.465, indicating that there is a significant difference between the means of the questions and the measure being analyzed. It is also worth noting that the variance is an indicator of the spread of the data; the larger the variance, the more spread out the data is, and it could involve the presence of outliers.

A low standard deviation means that most of the scores are near the mean, and a high value means that the scores are more dispersed. To identify which questions are considered more significant by the participants, the average scores were evaluated and contrasted among the questions. Questions with higher average scores are considered more significant by the participants. Furthermore, questions with a lower standard deviation imply that the responses are more consistent; hence, it is more likely that the question is considered more important by the participants. Based on the results from Table 4, in hierarchical order, starting with the most important, questions Q1, Q12, Q13, Q9, and Q10 are the most significant for the participants in terms of importance and consistency.

To determine which questions are not well understood, apart from the standard deviation, the distribution of responses was calculated and analyzed. The distribution of scores is a measure of how the scores are distributed across the range for each question. It can be visualized for all 20 questions using the histogram and the frequency distribution presented in Figure 1.

**Table 4.** Standard deviation and variance for the 20-question data set regarding public engagement.

| Question ID | Count | Sum | Average | Variance | Std. Dev. |
|---|---|---|---|---|---|
| Q1 | 44 | 211 | 4.795 | 3.701 | 1.924 |
| Q2 | 44 | 202 | 4.591 | 4.108 | 2.027 |
| Q3 | 44 | 176 | 4.000 | 4.047 | 2.012 |
| Q4 | 44 | 186 | 4.227 | 5.110 | 2.261 |
| Q5 | 44 | 183 | 4.159 | 4.928 | 2.220 |
| Q6 | 44 | 166 | 3.773 | 3.901 | 1.975 |
| Q7 | 44 | 168 | 3.818 | 3.966 | 1.992 |
| Q8 | 44 | 153 | 3.477 | 3.790 | 1.947 |
| Q9 | 44 | 189 | 4.295 | 4.120 | 2.030 |
| Q10 | 44 | 188 | 4.273 | 4.296 | 2.073 |
| Q11 | 44 | 176 | 4.000 | 4.419 | 2.102 |
| Q12 | 44 | 207 | 4.705 | 3.469 | 1.862 |
| Q13 | 44 | 204 | 4.636 | 3.958 | 1.989 |
| Q14 | 44 | 160 | 3.636 | 5.027 | 2.242 |
| Q15 | 44 | 177 | 4.023 | 5.465 | 2.338 |
| Q16 | 44 | 167 | 3.795 | 5.236 | 2.288 |
| Q17 | 44 | 192 | 4.364 | 4.423 | 2.103 |
| Q18 | 44 | 194 | 4.409 | 4.619 | 2.149 |
| Q19 | 44 | 171 | 3.886 | 4.615 | 2.148 |
| Q20 | 44 | 174 | 3.955 | 5.207 | 2.282 |

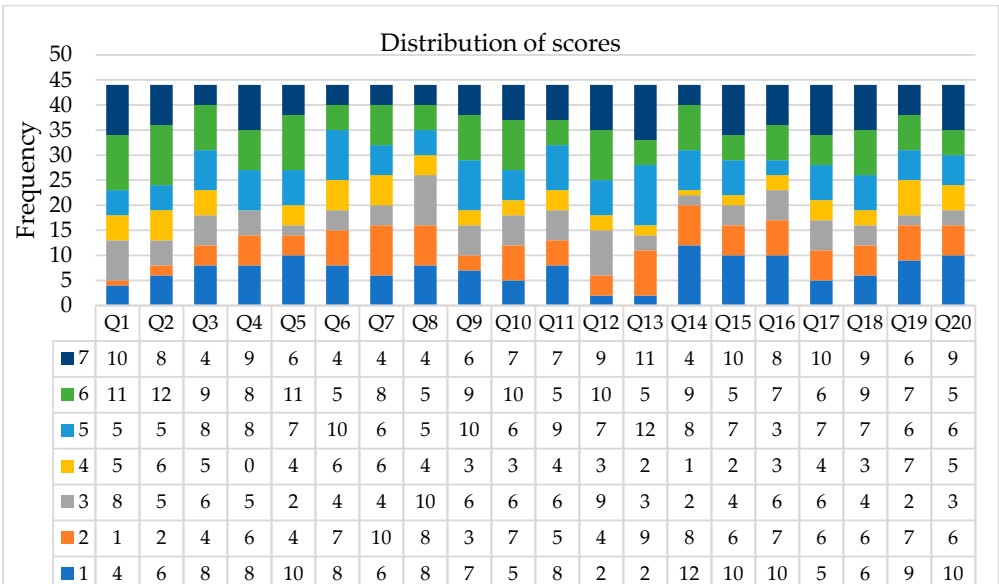

| | Q1 | Q2 | Q3 | Q4 | Q5 | Q6 | Q7 | Q8 | Q9 | Q10 | Q11 | Q12 | Q13 | Q14 | Q15 | Q16 | Q17 | Q18 | Q19 | Q20 |
|---|---|---|---|---|---|---|---|---|---|---|---|---|---|---|---|---|---|---|---|---|
| ■ 7 | 10 | 8 | 4 | 9 | 6 | 4 | 4 | 4 | 6 | 7 | 7 | 9 | 11 | 4 | 10 | 8 | 10 | 9 | 6 | 9 |
| ■ 6 | 11 | 12 | 9 | 8 | 11 | 5 | 8 | 5 | 9 | 10 | 5 | 10 | 5 | 9 | 5 | 7 | 6 | 9 | 7 | 5 |
| ■ 5 | 5 | 5 | 8 | 8 | 7 | 10 | 6 | 5 | 10 | 6 | 9 | 7 | 12 | 8 | 7 | 3 | 7 | 7 | 6 | 6 |
| ■ 4 | 5 | 6 | 5 | 0 | 4 | 6 | 6 | 4 | 3 | 3 | 4 | 3 | 2 | 1 | 2 | 3 | 4 | 3 | 7 | 5 |
| ■ 3 | 8 | 5 | 6 | 5 | 2 | 4 | 4 | 10 | 6 | 6 | 6 | 9 | 3 | 2 | 4 | 6 | 6 | 4 | 2 | 3 |
| ■ 2 | 1 | 2 | 4 | 6 | 4 | 7 | 10 | 8 | 3 | 7 | 5 | 4 | 9 | 8 | 6 | 7 | 6 | 6 | 7 | 6 |
| ■ 1 | 4 | 6 | 8 | 8 | 10 | 8 | 6 | 8 | 7 | 5 | 8 | 2 | 2 | 12 | 10 | 10 | 5 | 6 | 9 | 10 |

**Figure 1.** Distributions of scores for the public engagement data set.

For example, for question Q1, the frequency of scores is given by {1:4, 2:1, 3:8, 4:5, 5:5, 6:11, 7:10}. Four respondents gave a score of 1, one respondent gave a score of 2, eight respondents gave a score of 3, and so on. Questions with a wide range of responses and a high standard deviation are generally not well understood. For all 20 questions, the calculated range was 6. Although the standard deviation for all questions is low, the study requires further clarifications for question Q15. The average values for the question range from 3.477 to 4.795, with Q1 having the highest average value of 4.795. The participants generally agreed that the universities' commitment to public engagement is clearly defined. However, it is worth noting that the average for Q1 is only slightly above the midpoint of the scale (4.5), which means that the results are not overwhelmingly in favor of the statement. There were some participants who disagreed or were uncertain about the statement; thus; there is a need for further investigation [18].

Regarding the documentation of public commitment (Q2), the lowest results were recorded in Greece (with an average of 3.85) and the best results were recorded in Germany with an average of 6.33, indicating that German universities have the best practices for documentation of public engagement activities. The results suggest that the commitment to public engagement is well documented, but there may be room for improvement in terms of clarity and dissemination of information. As other research shows, confusion on the subject can be due to a lack of consistency in the channels of information and the diversity of tools [11,34]. In order to further investigate this issue, Q3 was analyzed.

According to the respondents, most universities make efforts so that their documented commitment to public engagement is known and understood; there are no significant differences between the partner countries. The conclusion aligns with several other findings at a European level and can be explained mainly due to cultural and societal similarities but also due to strategic collaboration paths between institutions [6,7,9,22,24]. Based on the results, it can be inferred that the universities may need to improve their efforts to ensure that their documented commitment to public engagement is also publicly known and understood. Such strategies are implemented and actively promoted by universities and institutions worldwide, but with notable differences in the effectiveness of the tools [26,33]. Depending on the cultural approach, universities need to establish the most effective methods for undertaking public engagement documentation.

When asked if people from different levels of the university are responsible for the implementation of the public involvement agenda (Q4), the respondents appreciated the efforts of the university staff, suggesting that there is a fair level of responsibility among people at different levels of the university for implementing the public engagement agenda. European universities tend to assume a high level of responsibility in undertaking academic third-mission actions, endeavors sustained by a variety of common efforts and initiatives [6,7,12,22]. However, there is still room for improvement as the mean score is not the highest, indicating that there may be some lack of clarity or understanding of the responsibilities related to public engagement across different levels of the university. Several studies found that lack of clarity can be due to improper communication throughout the universities' management and organizational hierarchies [17,19].

Surveyed universities are concerned with investments to encourage public involvement (average = 4.159 for Q5), but they are less involved in offering incentives and rewards to promote audience involvement (average = 3.773 for Q6). Some universities have been known to strongly encourage public engagement through student involvement, which has proven beneficial in the long-term development of third mission strategies [37]. The EU has promoted continuous development of public engagement through the academic third mission of universities [6], so as to counteract the gap between academia and entrepreneurs. The average score for Q6 is 3.773, which is relatively low compared to the other questions. For this question, the respondents generally disagree with or are neutral in their opinion that their universities offer incentives and rewards to promote public engagement. The standard deviation of 1.975 also infers that there is a significant amount of variation in the responses, indicating that some respondents may strongly disagree while others may be more neutral or slightly disagree. There is definitely room for improvement in this area for the universities in terms of offering incentives and rewards to promote public engagement. This is mainly performed through structural funds [8,9], but also through local initiatives [13,15].

The results for questions Q7, Q8, and Q9 were very close to the central tendency (average: Q7 = 3.818, Q8 = 3.477, Q9 = 4.295). Training activities to support public involvement are not sufficient, and services to promote public involvement are less satisfactory in surveyed universities. A fair interpretation of the obtained results could be that the respondents do not believe that the university is effectively integrating external services into its portfolio to promote public engagement. This was also the case for several other institutions outside of the study [15,20,21,30]. Thus, this is a clear area for improvement

for the university in terms of its public engagement efforts and is in correlation with other literature findings [32,39].

For questions Q10 and Q11 there are no significant differences between the results collected from different countries. These results reflect, in the opinion of the respondents, the satisfactory preoccupation of universities in using updated methods and approaches to develop public engagement skills among students and in the integration of public engagement practices in study programs [23]. The general opinion of the respondents is that they do not believe that the university is effectively integrating public engagement practices into its degree programs. For this question respondents stated that there are universities where the public is involved to some extent in the study programs. The justification for this statement is based, in the opinion of the respondents, on the fact that the universities consider the opinion of the public based on the feedback received from them, especially formulated during internships, and volunteering. It could be beneficial to follow up with strategies that have proven successful over one common framework [18,22,24].

By identifying the needs of external stakeholders (Q13 = 4.636), the universities are involved in the promotion of interdisciplinary educational paths (Q12 = 4.705), as the surveyed professors claim. Most of the participants think that their university is effectively promoting interdisciplinary educational paths. The results show that universities effectively promote interdisciplinary educational paths, and this is something that is positively perceived by the respondents, a result that aligns with most literature research [20,21,32].

Regarding the evaluation of the activities and results of public commitment (Q15 = 4.023) and indicators used (Q14 = 3.636), the best results were recorded in the universities of Romania and Lithuania, and lower results were obtained in Greece. These results could be explained by the fact that the respondents from Romania are teaching staff directly involved in the evaluation activity, compared to Greece, where doctoral students were involved in the survey. This context also explains the average obtained for question Q16 = 3.795 regarding the communication of the evaluation results on the impact of the institutions' activities. This issue is of particular importance in the process of standardization, and universities should address their challenges based on proven strategies [16]. Results suggest that the respondents feel that the universities are not effectively using indicators to measure their activities and public engagement results, and it may be beneficial for universities to review and improve their methods for measuring and evaluating the effectiveness of public engagement activities. Insight into these processes is given by literature and professionals [11,14,20]. The low average score and large variation in responses suggest that this may be an area where the university could improve in terms of public engagement efforts [2]. This set of data shows that there is a need for the universities to improve in integrating the results of their public engagement activities into future planning and organizational development [2,4]. The standard deviation of 2.103 for Q17 means that the responses to this question are relatively spread out. This is also supported by the distribution of scores. In the ANOVA table, the values reveal that there is a significant difference between the means of the different rows, inferring that the responses to this question vary between different groups. Regarding the influence of universities at the local and regional level in Q17, the lowest average was obtained for universities in Greece; for the other countries, the results were approximately equal.

Social impact from public involvement activities and the definition at the university level are not fully satisfactory for respondents from all countries (Q18, Q19), with the averages obtained being close to the recorded central tendency. This satisfactory result was also recorded for question Q20 regarding the integration of interested parties in the management of the institution. Based on the obtained results, it can be concluded that the universities are generally successful in setting and communicating the goals and objectives of their public engagement activities and have a clear sense of direction in terms of how they want to create impact. This is a positive indication and hints at the fact that the universities effectively communicate their purpose and objectives with regard to public engagement with their communities and stakeholders [13,15]. Relationships with various stakeholders are crucial for universities in order to train students for real-life case scenarios

and offer a smooth transition to the job market. Integration initiatives include joint labs, entrepreneurship accelerators, spin-off communities, and many others, for the mutual benefit of universities and companies alike [13,20,21,36,39].

In order to avoid the dependence between two quantitative variables in the sample of data collected by applying the questionnaire, Pearson's correlation coefficient (r) was determined. The obtained coefficients had values between –1 (perfectly negative correlation) and 1 (perfectly positive correlation). The sign of the coefficient represents the meaning of the correlation, namely: the positive value corresponds to the variations of the same meaning and the negative one to those of the opposite direction. The absolute values of the correlation coefficients, presented in Table 5, express the intensity of the association between the items. Thus, for $\alpha < 0.05$, values of the correlation coefficient from $-0.25$ to $0.25$ were obtained, representing a weak or zero correlation, from $0.25$ to $0.50$ (or from $-0.25$ to $-0.50$) acceptable degree of association, from $0.50$ to $0.75$ (or from $-0.50$ to $-0.75$) moderate to good correlation, and from $0.75$ to $1$ (or from $-0.75$ to $-1$) very good correlation.

**Table 5.** Correlation of coefficients.

| | Q1 | Q2 | Q3 | Q4 | Q5 | Q6 | Q7 | Q8 | Q9 | Q10 | Q11 | Q12 | Q13 | Q14 | Q15 | Q16 | Q17 | Q18 | Q19 | Q20 |
|---|---|---|---|---|---|---|---|---|---|---|---|---|---|---|---|---|---|---|---|---|
| Q1 | 1.00 | | | | | | | | | | | | | | | | | | | |
| Q2 | 0.73 | 1.00 | | | | | | | | | | | | | | | | | | |
| Q3 | 0.77 | 0.84 | 1.00 | | | | | | | | | | | | | | | | | |
| Q4 | 0.39 | 0.46 | 0.42 | 1.00 | | | | | | | | | | | | | | | | |
| Q5 | 0.49 | 0.55 | 0.61 | 0.66 | 1.00 | | | | | | | | | | | | | | | |
| Q6 | 0.47 | 0.53 | 0.64 | 0.65 | 0.74 | 1.00 | | | | | | | | | | | | | | |
| Q7 | 0.46 | 0.53 | 0.56 | 0.53 | 0.56 | 0.66 | 1.00 | | | | | | | | | | | | | |
| Q8 | 0.43 | 0.50 | 0.49 | 0.61 | 0.54 | 0.56 | 0.36 | 1.00 | | | | | | | | | | | | |
| Q9 | 0.46 | 0.50 | 0.59 | 0.48 | 0.67 | 0.72 | 0.55 | 0.62 | 1.00 | | | | | | | | | | | |
| Q10 | 0.46 | 0.53 | 0.63 | 0.55 | 0.59 | 0.62 | 0.46 | 0.49 | 0.71 | 1.00 | | | | | | | | | | |
| Q11 | 0.43 | 0.57 | 0.60 | 0.60 | 0.47 | 0.61 | 0.69 | 0.38 | 0.47 | 0.54 | 1.00 | | | | | | | | | |
| Q12 | 0.12 | 0.24 | 0.17 | 0.27 | 0.13 | 0.35 | 0.33 | 0.31 | 0.37 | 0.35 | 0.52 | 1.00 | | | | | | | | |
| Q13 | 0.22 | 0.37 | 0.30 | 0.40 | 0.46 | 0.56 | 0.57 | 0.36 | 0.56 | 0.37 | 0.59 | 0.65 | 1.00 | | | | | | | |
| Q14 | 0.31 | 0.54 | 0.44 | 0.51 | 0.49 | 0.55 | 0.52 | 0.51 | 0.47 | 0.35 | 0.62 | 0.60 | 0.67 | 1.00 | | | | | | |
| Q15 | 0.32 | 0.55 | 0.52 | 0.51 | 0.58 | 0.54 | 0.66 | 0.37 | 0.54 | 0.55 | 0.66 | 0.52 | 0.70 | 0.85 | 1.00 | | | | | |
| Q16 | 0.47 | 0.55 | 0.61 | 0.53 | 0.58 | 0.64 | 0.66 | 0.47 | 0.53 | 0.46 | 0.72 | 0.45 | 0.67 | 0.74 | 0.77 | 1.00 | | | | |
| Q17 | 0.48 | 0.35 | 0.40 | 0.44 | 0.50 | 0.61 | 0.57 | 0.41 | 0.47 | 0.35 | 0.41 | 0.26 | 0.48 | 0.40 | 0.48 | 0.53 | 1.00 | | | |
| Q18 | 0.59 | 0.51 | 0.60 | 0.53 | 0.64 | 0.64 | 0.71 | 0.41 | 0.58 | 0.58 | 0.64 | 0.34 | 0.54 | 0.62 | 0.76 | 0.74 | 0.69 | 1.00 | | |
| Q19 | 0.52 | 0.55 | 0.62 | 0.63 | 0.61 | 0.69 | 0.76 | 0.54 | 0.64 | 0.64 | 0.72 | 0.46 | 0.56 | 0.70 | 0.80 | 0.74 | 0.57 | 0.87 | 1.00 | |
| Q20 | 0.41 | 0.29 | 0.36 | 0.45 | 0.54 | 0.63 | 0.41 | 0.37 | 0.42 | 0.31 | 0.45 | 0.37 | 0.55 | 0.51 | 0.50 | 0.59 | 0.50 | 0.61 | 0.54 | 1.00 |

Among all the survey items in the first part of the questionnaire, only positive values were recorded that corresponded to variations of the same meaning. There are some moderate-to-strong positive relationships between the different questions. For example, Q2 and Q3 have a correlation coefficient of 0.84, indicating a strong positive relationship between the two questions.

Q4 and Q5 have a correlation coefficient of 0.66, indicating a moderately positive relationship between the two questions. Similarly, Q5 and Q6 have a correlation coefficient of 0.74, indicating a moderately positive relationship between the two questions. The highest association was recorded between items Q18 and Q19 (0.87), Q2 and Q3 (0.84), and Q15 and Q19 (0.80). However, it can also be seen that there are some weaker or no relationships between certain questions. For example, Q10 and Q14 have a correlation coefficient of 0.35, indicating a weak relationship between the two questions, and Q8 and Q17 have a correlation coefficient of 0.41, indicating a moderate relationship between the two questions.

The weakest correlation between items was recorded between items Q12 and Q1 (0.12), Q12 and Q5 (0.13), and Q12 and Q3 (0.17). These results suggest that there are moderate to strong positive relationships between some of the questions, indicating that the answers to

these questions may be related to one another. However, there are also some weaker or no relationships between certain questions, indicating that the answers to these questions may not be as related to one another. It is important to keep in mind that correlation does not imply causation, and further analysis would be needed to understand the underlying relationships between the variables.

### 3.2. Relative Importance of Community Engagement

The questionnaire was put together so that the answers reflect a different facet of community engagement in European universities. Questions do not overlap in information but rather offer a complementary vision on how universities integrate community engagement practice into their academic third missions. Thus, each question is viewed both as a separate entity, with its own value in the setting of the overall objective of the questionnaire, and as a puzzle piece in the development of transformative actions.

In this context, results obtained by ANOVA and Pearson's correlation showed that further analysis is necessary to substantiate the construction of a cohesive framework that could impact the decision-making process regarding community engagement in European universities.

Given the complexity of the analyzed issue, the Analytic Hierarchy Process (AHP) was applied to define the importance of each one of the 20 questions, respectively, as an underlying component of community engagement. The authors identified AHP as the most suitable method, attributing its effectiveness to its ability to minimize biases in the results of the decision-making process [40,41]. This approach necessitated a total of 190 pairwise comparisons among all 20 questions. In AHP, a consistency ratio below 10% is considered acceptable for maintaining result accuracy [42]. Goepel's AHP Online System facilitated the analysis [43].

A decision matrix needs to be put together, evaluating the importance of each question in relation to all others and the degree of that importance. The used AHP scale was: 1—Equal Importance, 3—Moderate Importance, 5—Strong Importance, 7—Very Strong Importance, 9—Extreme Importance (2, 4, 6, 8 values in-between). To set the values for each pair of questions, the calculated standard deviation (Table 4) was used.

There are two important steps in putting together the matrix, as follows: 1. Which question is more important than the other; 2. How much more important is one question than the other based on the AHP scale. The first step is straight-forward as the question with the lowest standard deviation is the most important of the two being compared.

The second step involves weighing the differences in standard deviation and spreading them across the 9-point scale. A square matrix is used to calculate the standard deviation differences (1).

$$\begin{array}{c} \\ Q_1 \\ Q_2 \\ Q_3 \\ \cdots \\ Q_i \\ \cdots \\ Q_{20} \end{array} \begin{array}{c} Q_1 \quad Q_2 \quad Q_3 \quad \cdots \quad Q_j \quad \cdots \quad Q_{20} \\ \begin{pmatrix} x_{11} & x_{12} & x_{13} & \cdots & x_{1j} & \cdots & x_{120} \\ x_{21} & x_{22} & x_{23} & \cdots & x_{2j} & \cdots & x_{220} \\ x_{31} & x_{32} & x_{33} & \cdots & x_{3j} & \cdots & x_{320} \\ \cdots & \cdots & \cdots & \cdots & \cdots & \cdots & \cdots \\ x_{i1} & x_{i2} & x_{i3} & \cdots & x_{ij} & \cdots & x_{i20} \\ \cdots & \cdots & \cdots & \cdots & \cdots & \cdots & \cdots \\ x_{201} & x_{202} & x_{203} & \cdots & x_{20j} & \cdots & x_{2020} \end{pmatrix} \end{array} \tag{1}$$

where $x_{ij}$ is the difference between the standard deviation of question $Q_i$ and the standard deviation of question $Q_j$. If $x_{ij}$ has a negative value, then Question $Q_i$ is more important than question $Q_j$. Based on the maximum absolute value amongst these differences, each question gets assigned a point on the AHP scale, according to the procedure shown in Table 6.

**Table 6.** Criteria to assign points on the AHP scale for each pairwise comparison.

| Points on the AHP Scale | Interval Range for $\left\|x_{ij}\right\|$ When Assigning Points on the AHP Scale * |
|---|---|
| 1 | 0 |
| 2 | $\left(0,\ \frac{max\left\|x_{ij}\right\|}{n-1}\left((n-8)+\frac{1}{2}\right)\right]$ |
| 3 | $\left(\frac{max\left\|x_{ij}\right\|}{n-1}\left(1+\frac{1}{2}\right),\ \frac{max\left\|x_{ij}\right\|}{n-1}\left((n-7)+\frac{1}{2}\right)\right]$ |
| 4 | $\left(\frac{max\left\|x_{ij}\right\|}{n-1}\left(1+\frac{1}{2}\right),\ \frac{max\left\|x_{ij}\right\|}{n-1}\left((n-6)+\frac{1}{2}\right)\right]$ |
| 5 | $\left(\frac{max\left\|x_{ij}\right\|}{n-1}\left(1+\frac{1}{2}\right),\ \frac{max\left\|x_{ij}\right\|}{n-1}\left((n-5)+\frac{1}{2}\right)\right]$ |
| 6 | $\left(\frac{max\left\|x_{ij}\right\|}{n-1}\left(1+\frac{1}{2}\right),\ \frac{max\left\|x_{ij}\right\|}{n-1}\left((n-4)+\frac{1}{2}\right)\right]$ |
| 7 | $\left(\frac{max\left\|x_{ij}\right\|}{n-1}\left(1+\frac{1}{2}\right),\ \frac{max\left\|x_{ij}\right\|}{n-1}\left((n-3)+\frac{1}{2}\right)\right]$ |
| 8 | $\left(\frac{max\left\|x_{ij}\right\|}{n-1}\left(1+\frac{1}{2}\right),\ \frac{max\left\|x_{ij}\right\|}{n-1}\left((n-2)+\frac{1}{2}\right)\right]$ |
| 9 | $\left(\frac{max\left\|x_{ij}\right\|}{n-1}\left(1+\frac{1}{2}\right),\ \frac{max\left\|x_{ij}\right\|}{n-1}\left((n-1)+\frac{1}{2}\right)\right]$ |

* $n = 9$, the maximum value on the AHP scale.

Using the criteria given in Table 6, 190 comparisons were made in pairs and an AHP decision matrix was put together (Figure 2a). The relative importance of each question was calculated based on the decision matrix, using the principal eigenvector solution with five iterations and a delta value of $4.7 \times 10^{-8}$. Each question's weight was assigned based on the priority in the AHP Ranking, as shown in Figure 2b.

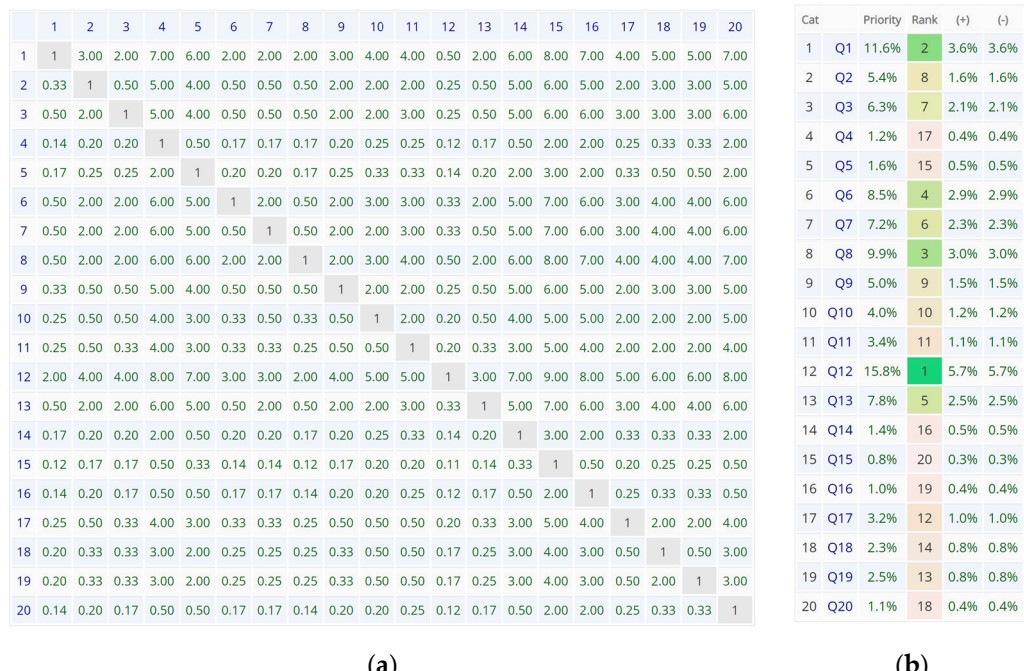

**Figure 2.** AHP for 20 questions on community engagement in European universities: (**a**) AHP Decision matrix; (**b**) AHP Ranking.

The consolidated results of the AHP reveal a consistency ratio of 3.5% (Figure 3), significantly lower than the predetermined threshold. Consequently, the model's inconsistencies

are within an acceptable range, allowing the derived importance coefficients to be reliably utilized in subsequent decisions.

Number of comparisons = 190
Consistency Ratio CR = 3.5%

Principal eigen value = 21.077
Eigenvector solution: 5 iterations, delta = 4.7E-8

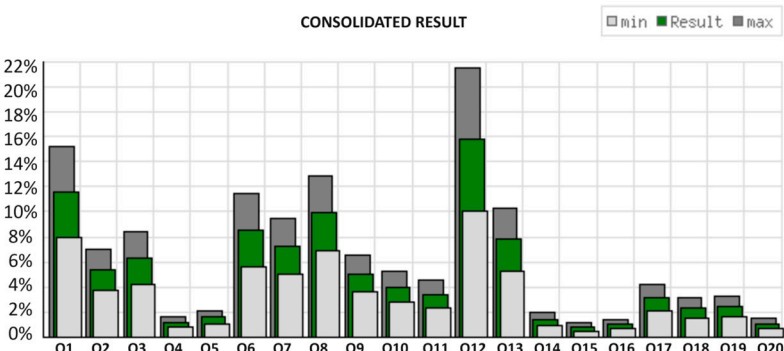

**Figure 3.** AHP consolidated result for all 20 questions on community engagement in European universities.

AHP shows that the most important questions relate to the promotion of interdisciplinary educational paths (Q12), the clarity of the public engagement definition (Q1), the integration of external services into universities' portfolios of services to promote public engagement (Q8), and the offer of incentives and rewards to promote public engagement (Q6). Q12, although the most important for the survey participant universities, has the lowest correlation coefficient of all questions, implying that this is a mandatory area of improvement and further investigation for all universities.

It is interesting to note that ANOVA identified Q1 as having the highest average value amongst the group, and according to AHP, it is the second most important component for universities. In this regard, there is a balance between value and importance, and further steps might involve improving functionality rather than value.

The ANOVA on Q8 showed that European universities do not effectively integrate external services into their portfolio to promote public engagement. This result corroborated its' importance. AHP shows that universities should implement a more efficient framework targeting practical solutions to external service integration. Q6 has strong positive values, with all other questions showing the grounded connection in research, making its' importance valuable for further analysis and improvement. Based on the AHP and ANOVA results the authors put together a set of recommendations and limitations fort the current study.

### 3.3. Recommendations and Study Limitations

The Academic Third Mission refers to the engagement of universities with their local communities through activities such as research, education, and services [5,23]. Public engagement, or the involvement of citizens in these activities, is crucial for the success of the Third Mission [35]. However, the results of the current study indicate that there are a number of challenges to effective public engagement in tertiary education. These challenges include a lack of awareness and understanding of the Third Mission among citizens, difficulty in involving citizens in decision-making processes, and conflicts of interest that arise in the participatory process. In light of these challenges, it is essential to develop strategies for improving public engagement in tertiary education through the Academic Third Mission [18,19,22]. Some possible strategies include increasing awareness and understanding of the Third Mission among citizens, involving citizens in decision-making processes and providing them with the tools and resources to participate effectively,

and addressing conflicts of interest in the participatory process. Based on the obtained results, the authors propose nine different strategies (S1 ÷ S9) for further development.

Improving public engagement in tertiary education requires a multifaceted approach, emphasizing transparency, early involvement, and a culture of participation. A key strategy is enhancing transparency and communication between universities and the community (S1). This can be effectively achieved by regularly publishing the results of participatory activities on the university's website and establishing a dedicated online channel to listen to and implement citizens' recommendations. Involvement of citizens should begin at the initial stages (S2), including the collection and processing of context data, identification of priorities, and planning and programming of interventions. Such early engagement ensures that their needs and perspectives are integral to decision-making processes. Additionally, fostering a culture of participation within the university is crucial (S3). This involves providing training and support to staff and students in participatory methods and encouraging active participation in decision-making processes. The formation of interest groups and coalitions during debates ensures diverse perspectives in decision-making (S4). Equally important is the regular evaluation and monitoring of the participation process (S5) to identify areas for improvement, ensuring inclusivity and fairness. Diverse participatory methods, such as town meetings, deliberative surveys, and design workshops, are essential to represent varied viewpoints (S6). Collaboration with other organizations and experts is another key aspect (S7), providing access to a broad range of perspectives and expertise in decision-making. It is also important to consider the available resources and the level of conflict (S8) related to the intervention area and the local community before implementing any strategy. Finally, supporting citizens to understand their needs and make informed decisions is paramount (S9). This includes informing them of the outcomes of the participatory process, the work conducted by researchers and experts, and collecting feedback for potential interventions and improvements. A specific online channel for listening to and implementing citizens' recommendations further supports this strategy, making for a more robust and inclusive approach to public engagement in tertiary education.

In order to facilitate the implementation of the above strategies, the study showed that there are still several areas in which universities can improve their engagement with citizens through the Academic Third Mission [1,4]. In order to effectively involve citizens in the decision-making process and ensure that their needs are being met, universities should consider implementing a variety of good practices. First, universities should prioritize transparency and communication throughout the participatory process. This includes clearly communicating the goals and objectives of the participatory process to citizens, as well as providing regular updates on the progress of the process and the outcomes achieved [2]. Universities should also make an effort to ensure that the results of the participatory process are widely shared and easily accessible to citizens, such as through a dedicated section on the university website. Second, universities should actively involve citizens in the planning and implementation of the Third Mission activities. This can be achieved through a variety of methods, such as working groups, town meetings, and participatory budgeting [20]. By involving citizens in the planning process, universities can ensure that their needs and priorities are taken into account and that the resulting interventions are more effective. Third, universities should consider providing support to citizens to understand their needs and make informed decisions. This can be achieved through a variety of methods, such as information desks, listening points, and providing information about the final result produced by the participatory process and the work conducted by researchers and experts [21,23,30]. Fourth, in order to prevent conflicts of interest, universities should have a clear policy in place to identify and address such situations. This can include the establishment of a conflict-of-interest committee, the implementation of a code of conduct, and the provision of training to staff and stakeholders on how to handle conflicts of interest [33,35]. Finally, universities should conduct regular evaluations of the participatory process to identify areas for improvement and ensure that the needs and priorities of citizens are being met. This can include conducting surveys or

focus groups to gather feedback from citizens, as well as conducting internal evaluations of the process [37].

The study revealed the main areas of improvement for the involved European universities and some important recommendations were proposed for further development. Based on these an initial framework is proposed in Figure 4.

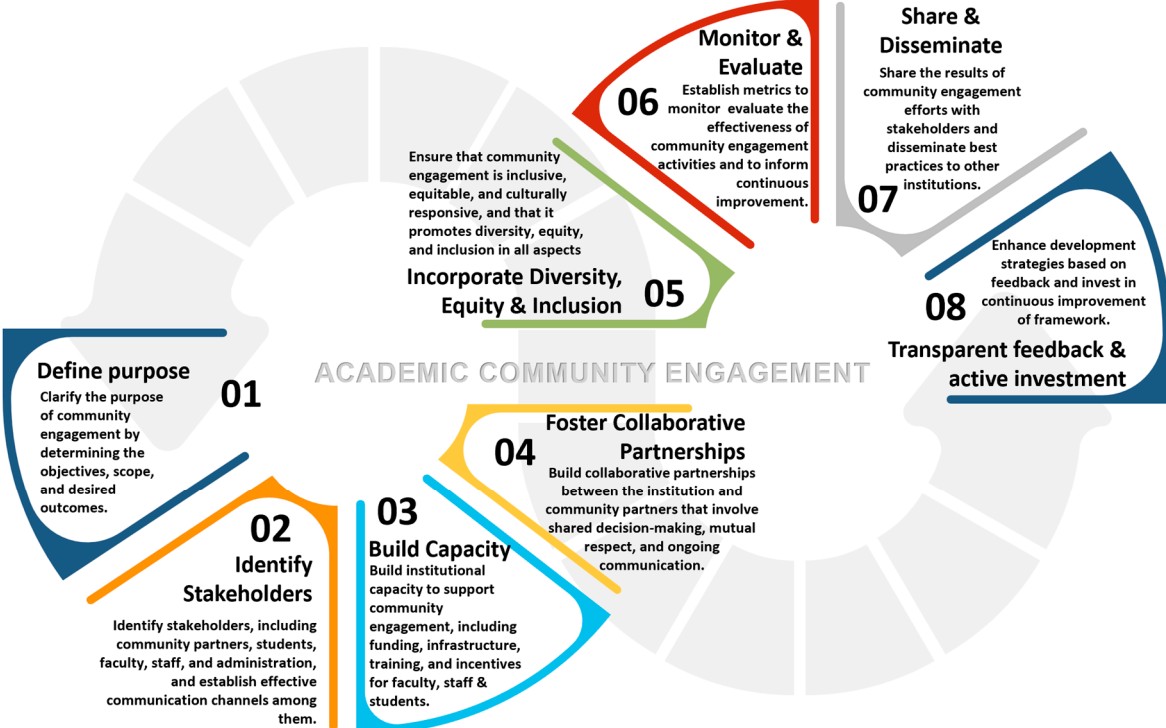

**Figure 4.** General framework for promoting academic community engagement.

To substantiate the framework and apply the identified sustainable strategies, the project consortium developed an online platform which enables stakeholders to get involved, participate and decide on sustainable academic contexts. The platform is available at www.tenacityplatform.com (accessed on 15 November 2023) and allows sustainable implementation of academic deliberative arenas for open science and innovation, and the delivery of an e-learning platform for academic deliberative practitioners. In accordance with study findings, the platform allows six main categories of stakeholders to participate in the creation of sustainable academic practices, namely: citizen, policy maker, professor, researcher student and teacher.

An important feature of this interactive tool is the iterative feedback loop which allows participants to the deliberative process to improve on any subroutine, enhancing the overall sustainability and probability of use for future applications. This approach also lowers the impact of identified limitations, all the way to potentially eliminating some of them. Multifunctionality was also promoted, and organic development of novel avenues was permitted, all leading to sustainable product development in academic settings.

Nevertheless, the study brings with it limitations which should be considered when assimilating the presented information and conclusions. One potential limitation of this study is the small sample size of the survey participants. With only 44 participants, it is difficult to generalize the findings to the larger population of citizens and universities. Small samples may have limited representativeness and statistical power, and assumptions such as normality can be more challenging to meet. Nonetheless, even a small quantitative study can establish baseline data on a topic, providing a starting point for future research and comparisons.

Additionally, the survey responses were self-reported and may not accurately reflect the true experiences and perspectives of the participants. The study also relies on the assumption that the participants have a clear understanding of the term "participatory practices" and have had similar experiences in their participation in university activities. There could also be a bias in the survey responses, as the participants may have had a vested interest in presenting their experiences in a certain way. Another limitation is that the study does not consider other factors that may influence the implementation of participatory practices in universities. For example, the survey does not take into account the specific political, economic, and cultural context of each university or the level of resources available to support participatory practices.

One mentionable limitation is that the study does not consider how the COVID-19 pandemic may have affected the ability of citizens and universities to participate in participatory practices, such as the shift to online engagement or the reduced availability of resources. The small sample size and self-reported nature of the survey responses, along with the assumptions made about the participants' understanding and experiences, may limit the generalizability of the findings. Also, the study does not take into account other factors that may influence the implementation of participatory practices in universities. To overcome the study limitations, it is recommended to conduct quantitative analysis and further research on larger studies. Future actions include the use of the current study as a pilot to inform a larger, more comprehensive research project. Additional qualitative methods, such as focus groups or case studies, will also supplement the survey data to provide a richer, more nuanced understanding of the third mission in different European HEIs, further developing the proposed framework.

The advantages of using ANOVA in our design analysis also counteract some of the study limitations. It allowed us to quantify trends and patterns for community engagement, even with the small sample size. This provided initial insights and identified potential areas of interest for further qualitative analysis. The quantitative data collection involved standardized instruments; the survey used Likert scales, allowing for consistency in data collection and facilitating comparisons across respondents and institutions.

## 4. Conclusions

The current study provides valuable insights into the current state of public engagement in tertiary education through the Academic Third Mission in European universities. The results of this survey can be used to identify gaps and areas for improvement in the development of strategies for promoting public engagement. Additionally, the study leads to the conclusion that European universities need a general framework for promoting and improving public engagement in tertiary education through the Academic Third Mission. Furthermore, the study's findings can be used to enrich a repository of good practices in Europe, which will be showcased in a handbook and on the TENACITY project website. This can serve as a valuable resource for universities looking to improve their public engagement strategies. The obtained results can be used to help identify the needs of universities in order to improve their deliberative practices. A survey was designed and applied to collect the data from 44 respondents, representing 35 universities from nine European countries. Quantitative (ANOVA) and qualitative analysis was undertaken to analyze various aspects of public engagement, such as university commitment, documentation, public awareness, investments, incentives, training, and stakeholder engagement.

The ANOVA results showed that while the respondents generally have a neutral opinion on the statements regarding public engagement at the university, there are some areas where they feel more positively or negatively. For example, the higher scores for Q1, Q2, and Q9 suggest that the respondents feel that the university's commitment to public engagement is clearly defined, well documented, and has well-structured target groups for its community public engagement activities. Lower scores for Q3, Q4, and Q5 show that the respondents feel that the university does not ensure that the documented commitment to public engagement is also publicly known and understood, people at different levels

of the university are not responsible for implementing the public engagement agenda, and the university does not currently make adequate investments to encourage public engagement. Similarly, higher scores for Q6 and Q7 imply that the respondents feel that the university offers incentives and rewards to promote public engagement and offers training activities to support public engagement. The smaller values obtained for Q8, Q10 and Q11 showcase the situation where the respondents feel that the university does not integrate external services into its portfolio of services to promote public engagement, does not use up-to-date methods and approaches to develop public engagement skills among students, and does not integrate public engagement practices into degree programs. Results for Q12, Q13 and Q19 were registered in the upper part of the evaluation scale and signify that the respondents think that the university promotes interdisciplinary educational paths, compares and identifies the needs of its external stakeholders, and has defined the kind of impact it aims to create through public engagement. On the other hand, lower scores for Q14, Q15 and Q16 suggest that the respondents feel that the university does not use indicators to measure its activities and public engagement results, does not ensure that the results of the impact assessment of public engagement activities are used for future planning and organizational development, and does not communicate the results of the assessment on the impact of its public engagement activities inside and outside the institution. Higher scores for Q17, Q18, and Q20 entail that the university influences community engagement at local and regional levels, creates a social impact from public engagement activities, and integrates community stakeholders into the institution's leadership.

AHP was used to add value to the current study by prioritizing the questions based on their relative importance, thus offering a comprehensive view that is beneficial for both analytical and decision-making purposes. The analysis identified four key survey areas: promoting interdisciplinary paths (Q12), defining public engagement (Q1), integrating external services (Q8), and incentivizing public engagement (Q6). Q12, crucial but with the lowest correlation, highlighted a significant improvement area. Q1's high average in ANOVA aligned with its AHP importance, suggesting a need to focus on functionality. Q8's poor integration of external services in universities, as per ANOVA, combined with its AHP significance, called for more efficient external service integration strategies. Q6's strong correlations indicated its vital role in research and improvement.

The current study is an important contribution to the field of public engagement in tertiary education through the Academic Third Mission by providing valuable insights and recommendations that can be used to improve the development of strategies and enhance public engagement in European universities.

**Author Contributions:** Conceptualization: M.-E.U. and C.-V.D.; Methodology: P.S.; Validation: C.-V.D., P.S. and M.-E.U.; Formal analysis: C.-V.D.; Investigation: M.-E.U.; Resources: P.S.; Data curation: P.S., M.-E.U. and C.-V.D.; Writing—original draft preparation: M.-E.U.; Writing—review and editing: P.S. and C.-V.D.; Visualization: M.-E.U.; Supervision: C.-V.D.; Project administration: P.S.; Funding acquisition: P.S. All authors have read and agreed to the published version of the manuscript.

**Funding:** This research was funded by the European Community's ERASMUS+ PROGRAMME under grant agreement no. 2021-1-IT02-KA220-HED-000032042—ACADEMIC THIRD MISSION: comuniTy Engagement for a kNowledge bAsed soCIeTY (TENACITY). This work was supported by a grant from the National Program for Research of the National Association of Technical Universities—GNAC ARUT 2023, Grant No. 14/06.10.2023.

**Institutional Review Board Statement:** Not applicable.

**Informed Consent Statement:** Not applicable.

**Data Availability Statement:** The data presented in this study are available in Appendix A.

**Acknowledgments:** Authors would like to acknowledge the contribution of University POLITEHNICA of Bucharest for the support given in the form of infrastructure and other resources not covered through the funded projects.

**Conflicts of Interest:** The authors declare no conflicts of interest. The funders had no role in the design of the study; in the collection, analyses, or interpretation of data, in the writing of the manuscript, or in the decision to publish the results.

## Appendix A

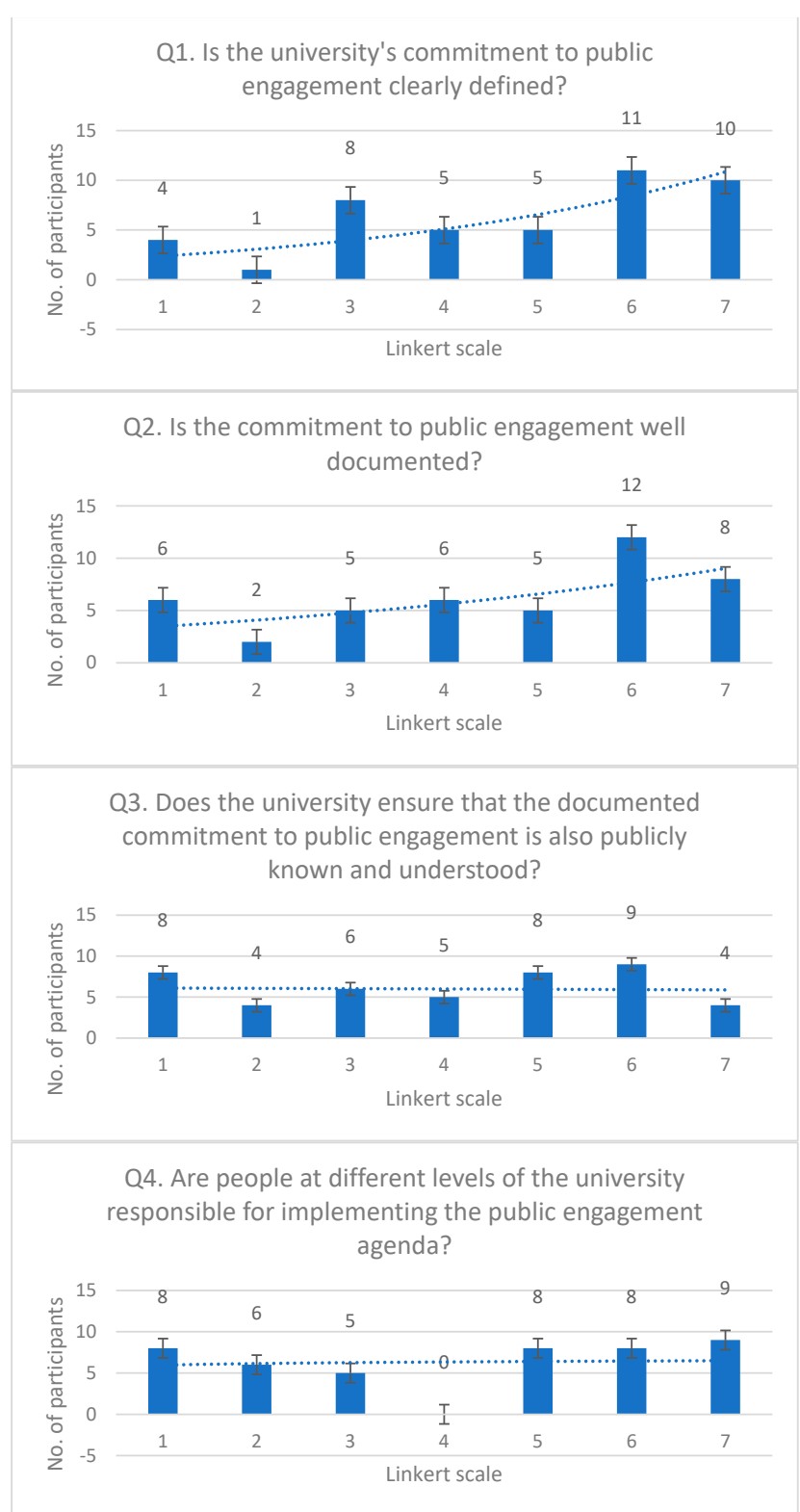

**Figure A1.** *Cont.*

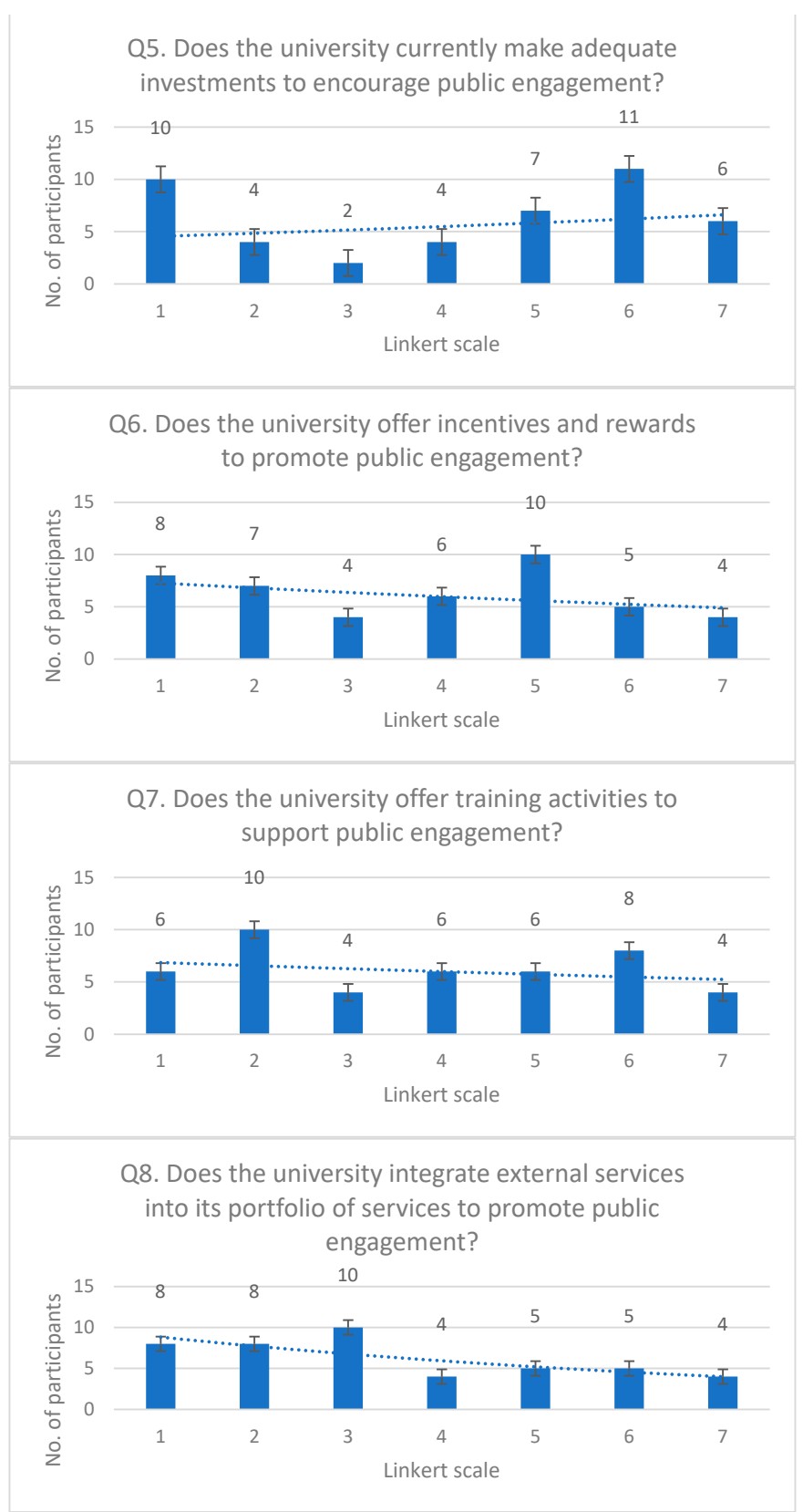

**Figure A1.** *Cont.*

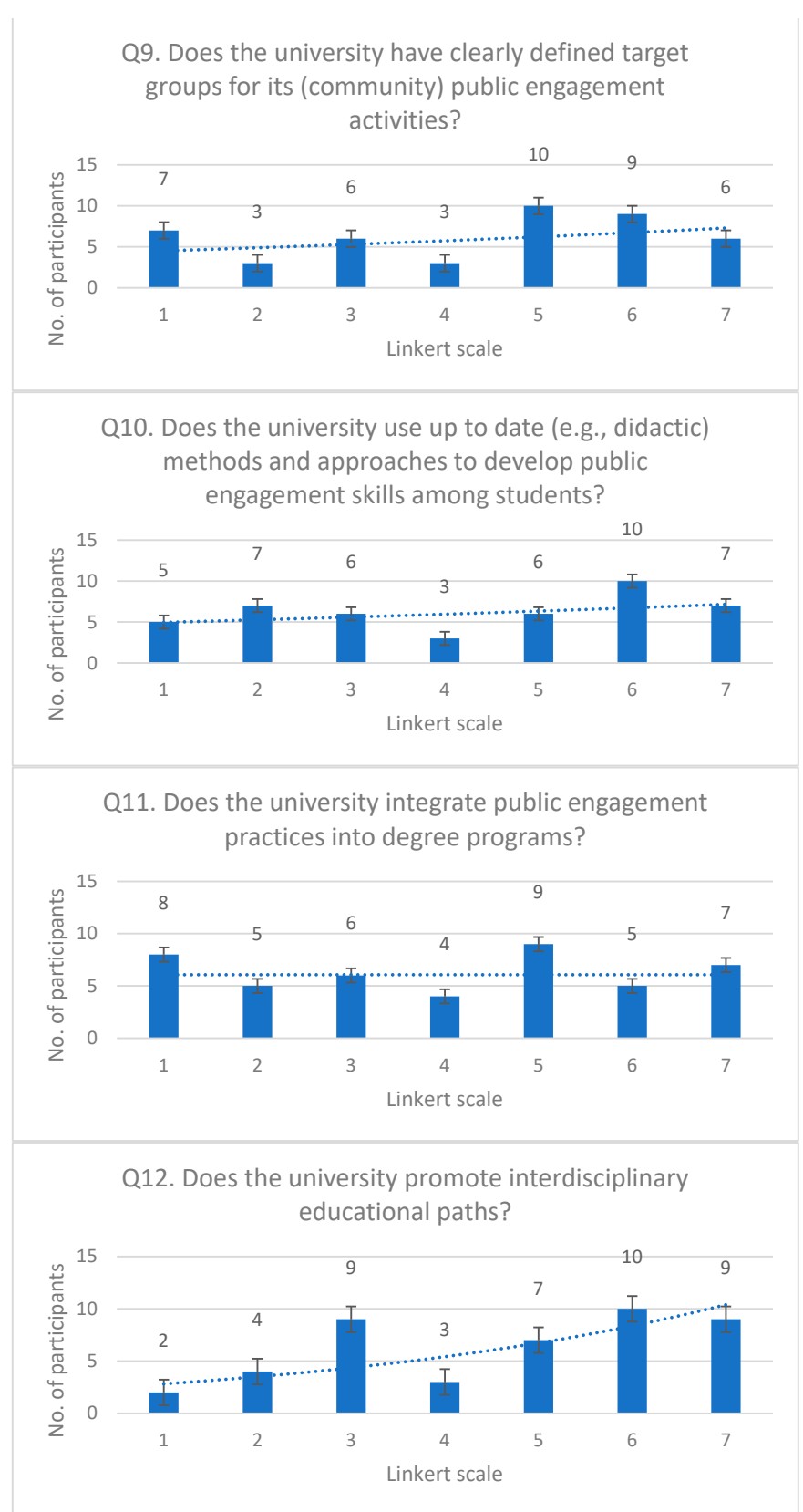

**Figure A1.** *Cont.*

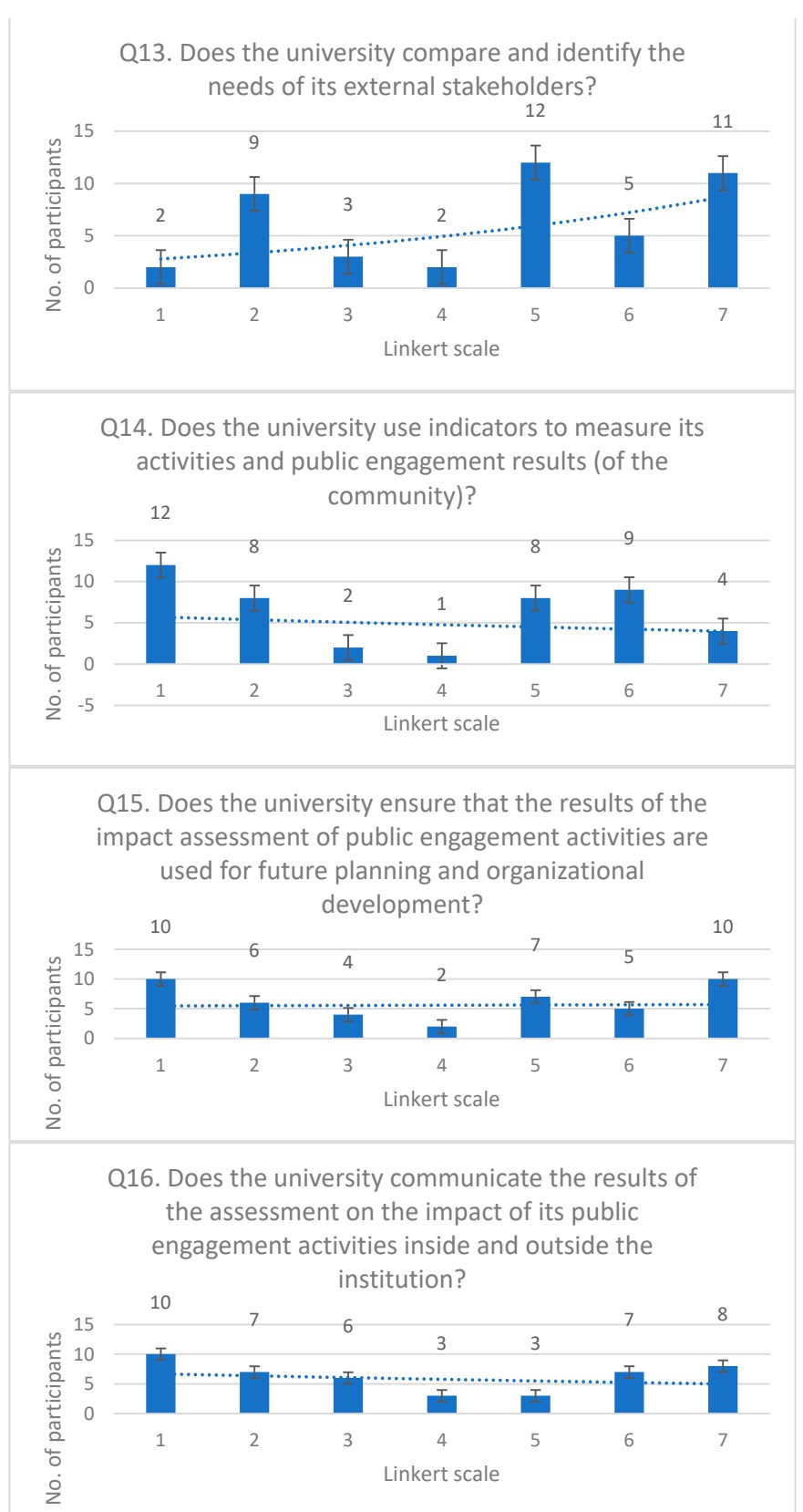

**Figure A1.** *Cont.*

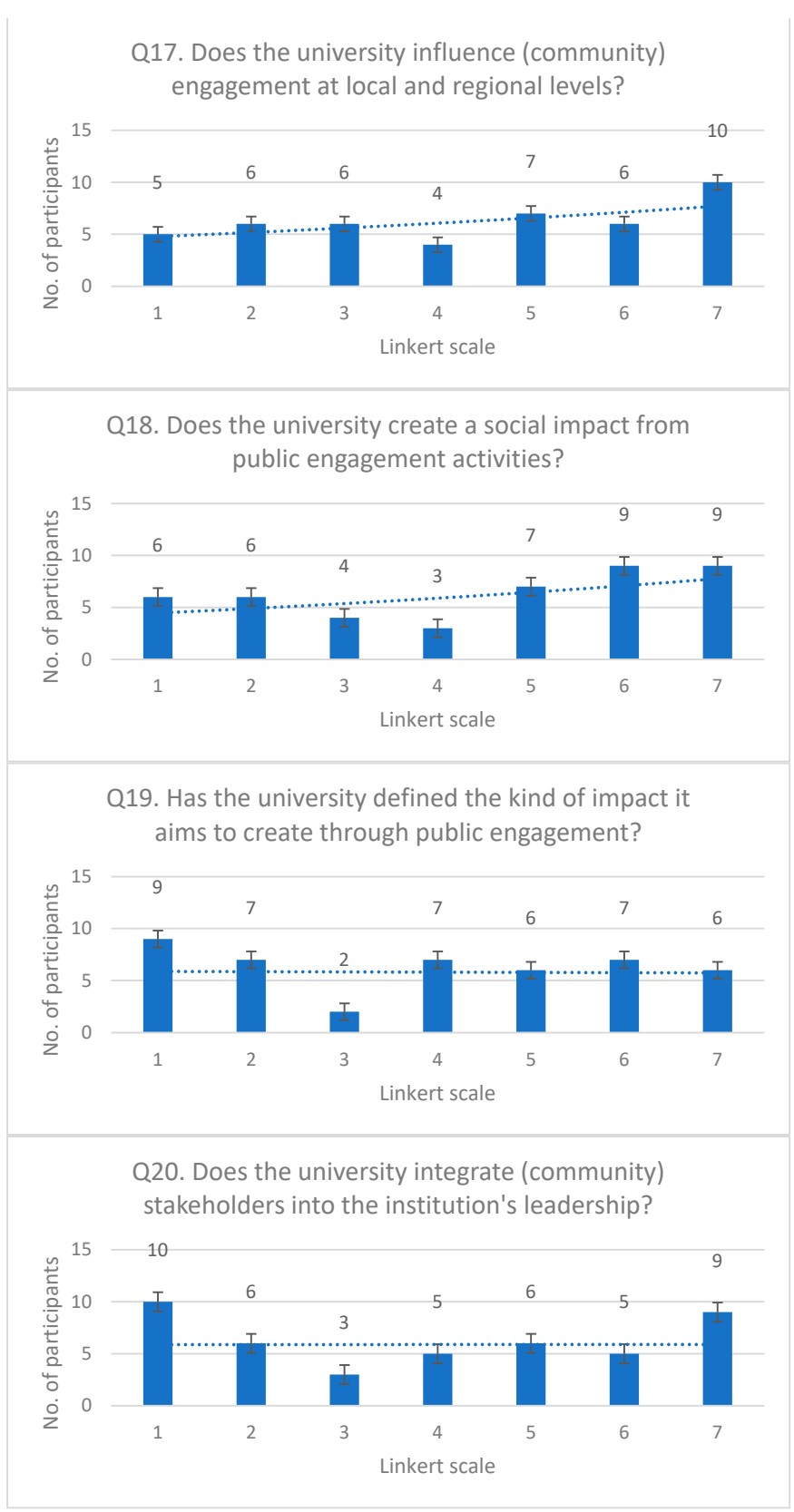

**Figure A1.** Results for the 20 items within TENACITY project.

## Appendix B

**Table A1.** Shapiro–Wilk test applied to calculate the statistic (W) and the *p*-value for each of the 20 questions from the survey in Germany, Greece and Lithuania.

| Question | Germany | | | Greece | | | Lithuania | | |
|---|---|---|---|---|---|---|---|---|---|
| | W | *p*-Value | Normality | W | *p*-Value | Normality | W | *p*-Value | Normality |
| Q1 | 0.629776 | 0.001241 | No | 0.94817 | 0.532673 | Yes | 0.860379 | 0.261574 | Yes |
| Q2 | 0.944664 | 0.682961 | Yes | 0.958862 | 0.704304 | Yes | 0.91099 | 0.487663 | Yes |
| Q3 | 0.849402 | 0.224231 | Yes | 0.899812 | 0.112078 | Yes | 0.971374 | 0.849971 | Yes |
| Q4 | 0.790653 | 0.086487 | Yes | 0.881089 | 0.060231 | Yes | 0.848079 | 0.219999 | Yes |
| Q5 | 0.944664 | 0.682961 | Yes | 0.819258 | 0.008724 | No | 0.894945 | 0.406387 | Yes |
| Q6 | 0.91099 | 0.487662 | Yes | 0.881597 | 0.061244 | Yes | 0.839702 | 0.194534 | Yes |
| Q7 | 0.863369 | 0.272453 | Yes | 0.859002 | 0.029495 | No | 0.963072 | 0.798227 | Yes |
| Q8 | 0.849402 | 0.224231 | Yes | 0.909098 | 0.152901 | Yes | 0.839702 | 0.194534 | Yes |
| Q9 | 0.992912 | 0.971877 | Yes | 0.845529 | 0.019323 | No | 0.992912 | 0.971878 | Yes |
| Q10 | 0.827427 | 0.161191 | Yes | 0.876281 | 0.051458 | Yes | 0.743573 | 0.033567 | No |
| Q11 | 0.629776 | 0.001241 | No | 0.934432 | 0.35164 | Yes | 0.863369 | 0.272453 | Yes |
| Q12 | 0.800563 | 0.103233 | Yes | 0.760175 | 0.001673 | No | 0.629776 | 0.001241 | No |
| Q13 | 0.93927 | 0.649878 | Yes | 0.904935 | 0.133024 | Yes | 0.848079 | 0.219999 | Yes |
| Q14 | 0.949706 | 0.714281 | Yes | 0.844588 | 0.018768 | No | 0.772907 | 0.061847 | Yes |
| Q15 | 0.827427 | 0.161191 | Yes | 0.857627 | 0.028237 | No | 0.763479 | 0.051229 | Yes |
| Q16 | 0.998396 | 0.995064 | Yes | 0.832679 | 0.013032 | No | 0.886912 | 0.369 | Yes |
| Q17 | 0.863369 | 0.272453 | Yes | 0.853856 | 0.025066 | No | 0.949706 | 0.714281 | Yes |
| Q18 | 0.944664 | 0.682961 | Yes | 0.900759 | 0.11568 | Yes | 0.949706 | 0.714281 | Yes |
| Q19 | 0.894945 | 0.406388 | Yes | 0.877539 | 0.053617 | Yes | 0.927082 | 0.577355 | Yes |
| Q20 | 0.927082 | 0.577355 | Yes | 0.856535 | 0.027278 | No | 0.629776 | 0.001241 | No |

**Table A2.** Shapiro–Wilk test applied to calculate the statistic (W) and the *p*-value for each of the 20 questions from the survey in Romania, Spain, Sweden.

| Question | Romania | | | Spain | | | Sweden | | |
|---|---|---|---|---|---|---|---|---|---|
| | W | *p*-Value | Normality | W | *p*-Value | Normality | W | *p*-Value | Normality |
| Q1 | 0.858486 | 0.146728 | Yes | 0.774708 | 0.022823 | No | 0.971374 | 0.849971 | Yes |
| Q2 | 0.858486 | 0.146728 | Yes | 0.813434 | 0.055481 | Yes | 0.949706 | 0.714281 | Yes |
| Q3 | 0.867412 | 0.176171 | Yes | 0.932528 | 0.572603 | Yes | 0.91099 | 0.487662 | Yes |
| Q4 | 0.846302 | 0.113659 | Yes | 0.784353 | 0.028585 | No | 0.894945 | 0.406387 | Yes |
| Q5 | 0.853883 | 0.133334 | Yes | 0.909711 | 0.393876 | Yes | 0.763479 | 0.051229 | Yes |
| Q6 | 0.929357 | 0.545445 | Yes | 0.926057 | 0.517886 | Yes | 0.949706 | 0.714281 | Yes |
| Q7 | 0.921579 | 0.481756 | Yes | 0.83571 | 0.090587 | Yes | 0.800563 | 0.103233 | Yes |
| Q8 | 0.910662 | 0.400475 | Yes | 0.879977 | 0.226348 | Yes | 0.728634 | 0.023857 | No |
| Q9 | 0.670536 | 0.001752 | No | 0.911128 | 0.403738 | Yes | 0.971374 | 0.849971 | Yes |
| Q10 | 0.719758 | 0.006067 | No | 0.955536 | 0.77965 | Yes | 0.882072 | 0.34756 | Yes |
| Q11 | 0.863961 | 0.164219 | Yes | 0.846302 | 0.113659 | Yes | 0.963072 | 0.798227 | Yes |

**Table A2.** *Cont.*

| Question | Romania | | | Spain | | | Sweden | | |
|---|---|---|---|---|---|---|---|---|---|
| | **W** | **_p_-Value** | **Normality** | **W** | **_p_-Value** | **Normality** | **W** | **_p_-Value** | **Normality** |
| Q12 | 0.840044 | 0.099451 | Yes | 0.907051 | 0.375833 | Yes | 0.882072 | 0.34756 | Yes |
| Q13 | 0.856091 | 0.139616 | Yes | 0.862486 | 0.159333 | Yes | 0.827427 | 0.16119 | Yes |
| Q14 | 0.871193 | 0.190135 | Yes | 0.874451 | 0.202933 | Yes | 0.743573 | 0.033567 | No |
| Q15 | 0.870328 | 0.186858 | Yes | 0.863961 | 0.164219 | Yes | 0.798526 | 0.099603 | Yes |
| Q16 | 0.863225 | 0.161763 | Yes | 0.812736 | 0.054621 | Yes | 0.882072 | 0.34756 | Yes |
| Q17 | 0.934584 | 0.590524 | Yes | 0.90903 | 0.389195 | Yes | 0.963072 | 0.798227 | Yes |
| Q18 | 0.834969 | 0.089147 | Yes | 0.945253 | 0.686389 | Yes | 0.882072 | 0.34756 | Yes |
| Q19 | 0.824948 | 0.071632 | Yes | 0.931918 | 0.567328 | Yes | 0.863369 | 0.272453 | Yes |
| Q20 | 0.791718 | 0.033888 | No | 0.965365 | 0.863218 | Yes | 0.839702 | 0.194534 | Yes |

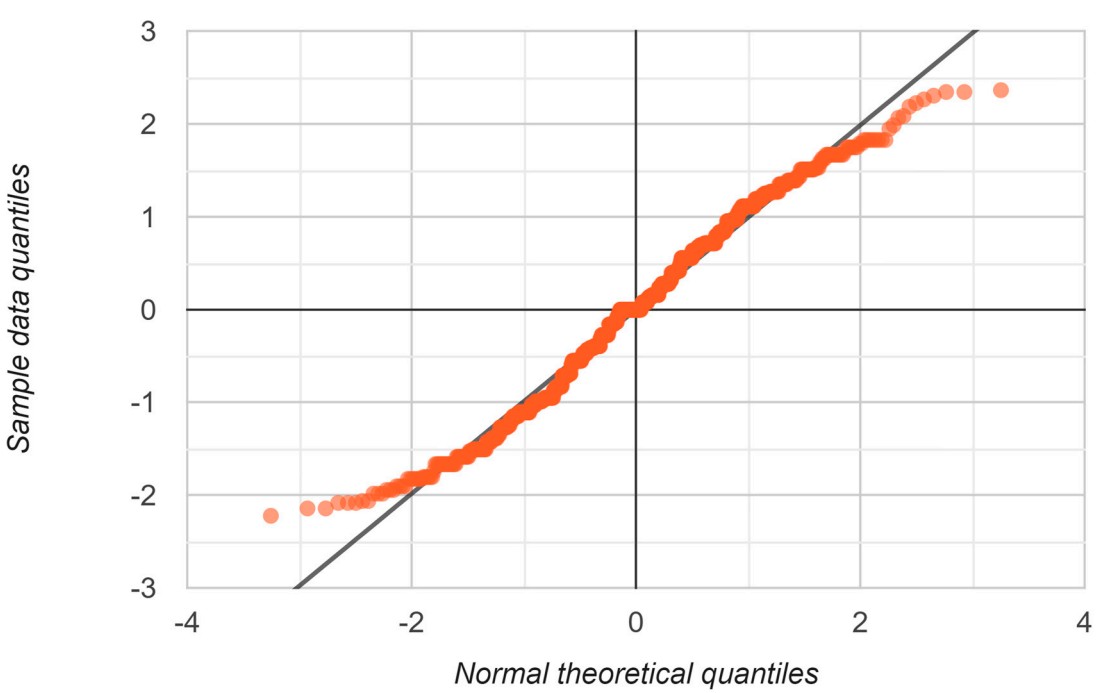

**Figure A2.** Q-Q plot of residuals.

**Appendix C**

**Table A3.** Levene's test for validation of homogeneity of variances for 20 questions of the survey (*p*-value > 0.05).

| | Spain | Romania | Italy | Sweden | Greece | Germany | Lithuania | Overall | Levene's Test Statistic | Levene's Test *p*-Value | Homogeneity |
|---|---|---|---|---|---|---|---|---|---|---|---|
| Q1 | 2.952381 | 6.619048 | 0.500000 | 2.916667 | 2.131868 | 0.250000 | 6.000000 | 3.721254 | 1.640097 | 0.165415 | Yes |
| Q2 | 2.571429 | 6.619048 | 0.000000 | 3.333333 | 3.362637 | 0.666667 | 5.666667 | 3.942509 | 1.725253 | 0.144158 | Yes |
| Q3 | 2.666667 | 6.238095 | 0.000000 | 5.666667 | 3.412088 | 2.250000 | 2.916667 | 3.997677 | 1.058790 | 0.405442 | Yes |

**Table A3.** *Cont.*

| | Spain | Romania | Italy | Sweden | Greece | Germany | Lithuania | Overall | Levene's Test Statistic | Levene's Test *p*-Value | Homogeneity |
|---|---|---|---|---|---|---|---|---|---|---|---|
| Q4 | 8.333333 | 6.666667 | 0.500000 | 6.333333 | 3.494505 | 3.583333 | 5.583333 | 5.027294 | 0.754531 | 0.610160 | Yes |
| Q5 | 5.904762 | 4.476190 | 0.500000 | 5.666667 | 3.758242 | 0.666667 | 1.583333 | 4.840883 | 0.829477 | 0.555211 | Yes |
| Q6 | 5.238095 | 2.904762 | 0.500000 | 3.333333 | 2.835165 | 5.666667 | 3.000000 | 3.865273 | 0.768821 | 0.599509 | Yes |
| Q7 | 4.238095 | 4.952381 | 2.000000 | 4.916667 | 2.527473 | 0.916667 | 4.916667 | 3.930314 | 0.367364 | 0.894622 | Yes |
| Q8 | 4.571429 | 5.285714 | 2.000000 | 8.333333 | 1.346154 | 2.250000 | 3.000000 | 3.816492 | 2.286003 | 0.057563 | Yes |
| Q9 | 3.619048 | 2.904762 | 0.000000 | 2.916667 | 3.609890 | 1.666667 | 6.666667 | 3.983740 | 1.494975 | 0.208598 | Yes |
| Q10 | 4.000000 | 1.810000 | 0.500000 | 8.667000 | 2.951000 | 2.000000 | 8.250000 | 4.063000 | 1.023930 | 0.426171 | Yes |
| Q11 | 6.670000 | 5.570000 | 0.500000 | 4.920000 | 3.450000 | 4.000000 | 8.250000 | 4.320000 | 0.614851 | 0.716864 | Yes |
| Q12 | 3.905000 | 1.905000 | 0.500000 | 8.667000 | 2.374000 | 4.917000 | 6.250000 | 3.503000 | 0.533000 | 0.884000 | Yes |
| Q13 | 3.619000 | 2.905000 | 2.000000 | 2.000000 | 3.346000 | 7.583000 | 5.583000 | 4.007000 | 0.604000 | 0.725000 | Yes |
| Q14 | 6.238000 | 6.952000 | 0.000000 | 4.667000 | 6.527000 | 2.333000 | 8.333000 | 5.928000 | 0.559000 | 0.784000 | Yes |
| Q15 | 5.571429 | 5.285714 | 0.500000 | 10.250000 | 4.686813 | 2.000000 | 5.666667 | 5.292102 | 0.781004 | 0.590487 | Yes |
| Q16 | 5.238100 | 6.904800 | 8.000000 | 8.666700 | 4.131900 | 4.333300 | 6.916700 | 5.356600 | 0.403400 | 0.871700 | Yes |
| Q17 | 5.619048 | 2.238095 | 2.000000 | 4.916667 | 3.456044 | 0.916667 | 3.333333 | 4.192799 | 0.622752 | 0.710774 | Yes |
| Q18 | 4.570000 | 5.810000 | 2.000000 | 8.670000 | 4.070000 | 0.670000 | 3.330000 | 4.670000 | 0.864975 | 0.530078 | Yes |
| Q19 | 3.571429 | 5.238095 | 2.000000 | 8.250000 | 3.719780 | 1.583333 | 4.666667 | 4.527294 | 0.395607 | 0.876791 | Yes |
| Q20 | 4.476190 | 7.476190 | 0.000000 | 3.000000 | 3.719780 | 4.666667 | 2.250000 | 4.987224 | 1.096671 | 0.383788 | Yes |

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
