# Peer review of "Academic Third Mission through Community Engagement: An Empirical Study in European Universities"

_education, doi:10.3390/educsci14020141_

Round 1

Reviewer 1 Report

Comments and Suggestions for Authors

In general, I acknowledge the potential of this paper; however, it is crucial to address certain limitations within the study.

A central concern revolves around the persuasiveness of the evaluation and validation methods employed in the proposed approach. It is imperative to conduct a comprehensive investigation into the prerequisites associated with the applied method.

Specifically, the paper lacks an explanation for the selection of ANOVA and whether the prerequisites for its application have been duly met. Furthermore, the utilization of Cronbach's alpha necessitates careful consideration of prerequisites, and the limitations arising from very small sample sizes are overlooked. The rationale for not exploring other well-established methods for analyzing Likert scales, such as PCA and factor analysis, remains unclear.

Addressing these concerns will enhance the overall rigor and credibility of the paper.

Comments on the Quality of English Language

The proficiency of the English language in the paper appears satisfactory as far as my assessment goes.

Author Response

Dear Reviewer,

Thank you for evaluating our manuscript and for the suggestions you gave to improved it. We attach a complete reply to your comments. 

Reviewer 2 Report

Comments and Suggestions for Authors

This is a well-written paper that has a theoretical contribution to the scholarship of engagement. All aspects of engagement have been covered in the validated questionnaire. This in itself is the methodological contribution of this paper to the scholarship of engagement.  

Author Response

Dear Reviewer,

Thank you for evaluating our manuscript and we appreciate your valuable input. 

Reviewer 3 Report

Comments and Suggestions for Authors

The article presents a quantitative study on Academic Third Mission through  community engagement. The study is reasonable in terms of pointing out the political relevance of the issue. In addition, there could be more emphasis on contextualizing the study scientifically, this especially with regard to theoretical discussions on the changing role of universities and to existing empirical research on evaluating/measuring the Third Mission. The innovative character of the study is indicated, but could be further underlined e.g. by pointing out the advantage of applying quantitative methods even if the sample is relatively small. This would also help to justify the (re-)construction of the "general" framework for promoting community engagement (page 15). Overall, the article is a valuable contribution to further discourse and research. 

Author Response

Dear Reviewer,

Thank you for evaluating our manuscript and for the suggestions you gave to improved it. We agree with your comments and have made the necessary improvements for pointing out the advantages of ANOVA (lines 630-633, 652-662). We also analysed the existence of prerequisites for applying ANOVA (lines 144 – 176, lines 179 – 212, lines 750-762) contributing to the rigor of evaluating Third Mission constructs in relation to existing approaches.